# Electrical Conductive Properties of 3D-PrintedConcrete Composite with Carbon Nanofibers

**DOI:** 10.3390/nano12223939

**Published:** 2022-11-08

**Authors:** Guido Goracci, David M. Salgado, Juan J. Gaitero, Jorge S. Dolado

**Affiliations:** 1Centro de Física de Materiales, CSIC-UPV/EHU, Paseo Manuel de Lardizábal 5, 20018 Donostia-San Sebastián, Spain; 2TECNALIA, Basque Research and Technology Alliance (BRTA) Parque Tecnológico de Bizkaia, Astondo Bidea, 48160 Derio, Spain; 3Donostia International Physics Center, Paseo Manuel de Lardizabal 4, 20018 Donostia-San Sebastián, Spain

**Keywords:** electrically conductive, 3D printing, CNFs, cement composites, smart materials

## Abstract

Electrical conductive properties in cement-based materials have received attention in recent years due to their key role in many innovative application (i.e., energy harvesting, deicing systems, electromagnetic shielding, and self-health monitoring). In this work, we explore the use 3D printing as an alternative method for the preparation of electrical conductive concretes. With this aim, the conductive performance of cement composites with carbon nanofibers (0, 1, 2.5, and 4 wt%) was explored by means of a combination of thermogravimetric analysis (TGA) and dielectric spectroscopy (DS) and compared with that of specimens prepared with the traditional mold method. The combination of TGA and DS gave us a unique insight into the electrical conductive properties, measuring the specimens’ performance while monitoring the amount in water confined in the porous network. Experimental evidence of an additional contribution to the electrical conductivity due to sample preparation is provided. In particular, in this work, a strong correlation between water molecules in interconnected pores and the σ(ω) values is shown, originating, mainly, from the use of the 3D printing technique.

## 1. Introduction

In recent years, much attention has been paid to the development of smart and multifunctional cement-based materials to meet the requirements of improving advanced engineering infrastructure [1,2]. For instance, several studies focused on innovative systems of energy harvesting through thermometric concretes [3,4,5,6,7], deicing [8,9,10,11,12,13], electromagnetic shielding [14,15,16,17,18,19,20,21,22,23,24], and health monitoring [25,26,27,28,29]. All these applications required not only the improvement of the mechanical properties of the cement-based concrete, but also the enhancement of secondary properties such as electrical conductivity to adapt the composite to the desired function. In this scenario, it is a major challenge to obtain a conductive cement-based material as the electrical resistivity of cement is in the range of insulators. The most common method to improve electrical conductivity is to incorporate conductive fillers into the cement matrix. Various forms (particles, fibers) and sizes (from nano to macro) of these additives have been used. In terms of the type of filler, particular attention has been paid to carbonaceous nanomaterials such as carbon nanofibers (CNFs) [30,31,32,33], carbon nanotubes (CNTs) [30,32,34,35,36], graphene nanoplatelets (GNPs) [27,37,38], as well as carbon black (CB) [37,39] due to both their high intrinsic conductivity and their good dispersion in the matrix. So far, it is possible to obtain electrically conductive concrete with conductivity in the range of 1 × 10^−5^–1 × 10^−2^ S/cm. Such a broad window reflects the complexity of the topic. Indeed, the results depend not only on the type and dosage of the filler, but also on the water-to-cement ratio, the mixing procedure, and the dispersion of the conductive filler. In this paper, we would like to contribute further to the study of electrical conductivity and explore how 3D concrete printing (3DCP) can influence the conductive properties of concrete composites. 3DCP has been under development for a decade and has made rapid progress in recent years [40,41,42]. However, so far, research on new concrete mixes for 3DCP has mainly focused on building construction. Therefore, the research aims to improve the mechanical properties of the 3D-printed material [43,44,45] and the hardened state [46,47,48] of the paste. However, the pumping and extrusion processes that take place during 3D printing could have a strong impact on other properties such as the electrical conductivity of cement composites. In fact, the use of an extruder can promote the dispersion of the conductive filler and the alignment of the conductive fibers due to the shear forces that occur [49].

To this end, this paper investigated the effect of the extrusion process in cement composites with different dosages (1, 2.5, and 4 wt%) of carbon nanofibers (CNFs). The amount of conductive filler was chosen to ensure a complete conductive pathway, as it has been observed in previous work that the percolation threshold for CNFs in cement composites is between 0.5 and 1 wt% [50,51,52,53,54]. In the first part of the manuscript, a comparison of the conductive performance of samples prepared with the traditional molding process and the 3D printing process is made. Since the CNFs in the 3D-printed cement composites do not appear to be the only major factor affecting the electrical conductivity at room temperature, the second part of this paper discusses the results of the experimental combination of dielectric spectroscopy and thermogravimetric analysis as a function of temperature. The aim of this experimental setup is to measure the electrical conductivity (σ(ω)) while monitoring the water content in the samples. It was found that free water molecules in the porous network modified by the 3D printing process contribute greatly to the conductivity.

## 2. Materials and Methods

The samples were prepared with ordinary Portland cement Cem I-52.5R, limestone aggregate with a particle size of less than 2 mm, distilled water (w/c = 0.6), and a carbon nanofiber dispersion (4 wt%) provided by Antolin group. The exact composition and reference name of all samples are presented in Table 1. The mixing procedure consisted of 30 s of slow mixing (according to EN 196-1) of all solids, followed by the addition of the carbon fiber dispersion and the remaining mixing water, if any, and mixing according to EN 196-1. The samples were prepared in two different ways. In the first case, the material was simply poured into 2 cm-wide prismatic molds. In the second case, the material was pressed into a 1 m-long and 2 cm-wide prismatic pipe with a ram extruder. In both cases, the specimens were demolded after 24 h and cured for 14 days at RH = 100 relative humidity. Then, the specimens were cut into smaller pieces (h ∼8 mm) with a diamond saw and cured for another week.

Thermogravimetric analyses (TGAs) were performed using a TGA-500 (TA Instruments, New Castle, DE, USA) to investigate the water content of the samples. All measurements were carried out under high-purity nitrogen flow at a temperature of 30–250 ${}^{\circ }C$ with a ramp rate of 1 ${}^{\circ }C$/min. Electrical conductivity was determined by measuring the complex impedance with a broadband dielectric spectrometer (DS), Novocontrol Alpha-A (Novocontrol, Montabaur, Germany) over the frequency range from 0.01 Hz to 1 MHz. Specimens (∼8 mm thickness) were placed between two gold-plated electrodes (diameter 10 mm). Before the measurement, the samples were kept at room temperature in the spectrometer for 10 min with a nitrogen gas flow to eliminate signals originating from moisture on the sample surface. One sample was examined for each dosage selected always from the same position of the larger extruded specimen. To improve the correlation between TGA and DS, a few milligrams of the sample were taken to be examined by thermogravimetric analysis. Both experiments were conducted in the same 24 h.

## 3. Results

To improve the readability of the manuscript, the work is divided into two sections. The first part (“Conductive Performance Comparison: Traditional Process vs. 3D Printing”) is dedicated to the comparison of the electrically conductive properties of mold-cast and extruded samples, while the second part (“Water’s Role in Electrical Conduction in 3D-Printed Cement Composites”) focuses on clarifying how extrusion affects the electrical conductivity of the cement composite and, in particular, the importance of the different water populations trapped in the cementitious matrix for the conduction mechanism.

### 3.1. Conductive Performance Comparison: Traditional Process vs. 3D Printing

In Figure 1, comparison is shown between the electrical conductivity σ (measured at room temperature at 100 Hz along the longitudinal direction of extrusion) of the sample produced by the traditional method and the 3D-printed composite. A different dependence on the amount of carbon filler can clearly be seen. The “traditionally” produced sample is characterized by a linear dependence on the CNF wt%. This result underlines the strong correlation between the electrical conductivity and the conductive filler in the samples produced by the traditional casting process. On the other hand, such dependence is lost when the samples are extruded (right side). A strong improvement in conductivity (+203%) is observed when the 3D printing process is used to produce the reference sample (0 wt% of CNFs). However, this improvement decreases with the amount of conductive filler. In particular, the sample printed with 4 wt% CNFs shows lower conductivity than the sample produced by the traditional casting method and is close to the conductivity measured for the 3D-printed sample without conductive fillers. The lack of the linear dependence of conductivity on the CNF dosage could indicate that there is an additional contribution to electrical conductivity from the extrusion process. This contribution is apparently stronger when no conductive fillers are used in the compound and decreases with the CNF dosage. To investigate whether this unique behavior is related to the orientation of the carbon nanofibers in the 3D-printed composites, conductivity was also measured along the transverse direction of extrusion. Indeed, if carbon nanofibers were randomly distributed in the specimen, an anisotropic behavior should be observed. On the contrary, when the CNFs are aligned in a preferred direction, a difference between σ(ω) measured along the transverse and longitudinal sections of the sample is detected. Figure 2 shows the percentage relationship between the conductivity measured in the longitudinal and transverse directions. For the sample with 2.5 wt%, conductivity measured in the longitudinal direction of extrusion is only ∼30% higher than that measured in the transverse direction. This result could indicate that only weak and partial alignment of carbon nanofibers is achieved in this cement composite. When we increased the dosage of CNF, an improvement in alignment was observed. Indeed, for an extruded sample with 4 wt% of CNFs, the electrical conductivity measured in the longitudinal direction is 80% higher than that measured in the transverse direction. Nevertheless, this sample is characterized by a worse conductivity than the sample with 2.5 wt% (see Figure 1). Consequently, these results suggest that, in our work, the alignment of carbon nanofibers is not followed by an improvement of the conductivity.

### 3.2. Water’s Role in Electrical Conduction in 3D-Printed Cement Composites

In this section, we focus on the effect of extrusion on the conductive properties of cement composites and their relationship to the water populations trapped in the porous network. Since water plays a very important role in the conductive properties, the extruded samples with 0 wt%, 1 wt%, 2.5 wt%, and 4 wt% were stored at relative humidity RH = 77 for 7 days to obtain the same moisture content in all samples. Subsequently, thermogravimetric measurements (TGA) were carried out up to 250 ${}^{\circ }C$. In this temperature range, we expected neither relevant changes in the structure of the sample, nor reactions leading to a phase transition. The variations in mass were only attributed to the loss of water molecules. Since the evaporation temperature depends on (a) the interaction between the water molecules and the surface host and (b) the size of the pore in which the water molecules are confined, TGA allows the identification and quantification of the water population in the cementitious porous network. Finally, the samples were investigated by dielectric spectroscopy to measure the conductivity from RT to 160 ${}^{\circ }C$. The same heating ramp (1 ${}^{\circ }C$/min) was used as in the TGA experiments to observe how the electrical properties vary as a function of temperature and, consequently, of the amount and type of water in the sample.

In Figure 3, the TGA response of specimens at different CNF dosages is shown. Each sample shows a different weight loss profile, suggesting a different water distribution in the porous network. The DTGA curve of the reference samples is characterized by a very broad peak centered at about 95 ${}^{\circ }C$. With the addition of CNFs (1 wt%), a shift of this peak towards a lower temperature is observed. Such a shift of the maximum is greater the higher the dosage of the conductive filler in the cement composite. Moreover, for the sample with 2.5 wt% of CNFs, we clearly detected an asymmetric derivative curve of the weight loss with two maxima centered at ∼60 ${}^{\circ }C$ and ∼95 ${}^{\circ }C$. These results indicate the presence of two main bulk water populations in the cement composites [55]. To quantify the different water populations in the samples, the temperature range was divided into four ranges according to the TGA results. The water content associated with each mass loss peak was calculated as $c_{w}=\left(w_{\Delta T}\right)/w_{250}$, where $w_{\Delta T}$ denotes the weight difference in the temperature range considered and $w_{250}$ denotes the weight at 250 ${}^{\circ }C$. Following the DTGA results, four temperature ranges were considered: Range I (T = 25–80 ${}^{\circ }C$) and Range II (T = 80–105 ${}^{\circ }C$) corresponding to bulk water; Range III (T = 105–160 ${}^{\circ }C$) and Range IV (T = 160–250 ${}^{\circ }C$) corresponding to the temperature range where trapped and structural water, respectively, are expected. The content of each water population is shown in Table 2.

Figure 4 shows the real part of the complex conductivity function as a function of temperature. The same pattern can be seen for all samples: (i) increase in conductivity (up to T = 80–90 ${}^{\circ }C$ (ii) plateau, and (iii) decrease in conductivity at T > 110 ${}^{\circ }C$. However, the extent of such a decrease in conductivity differed among the samples.

## 4. Discussion

In the first part of this paper, an additional contribution to electrical conductivity was observed in 3D-printed samples. By measuring the difference in σ(ω) between the longitudinal and transverse directions of the extruder, it was found that better alignment of the carbon nanofibers was not accompanied by higher electrical conductivity. Since water plays a fundamental role in the conductive properties of cementitious materials, a combination of TGA and DS measurements was used to investigate whether such an additional contribution could be related to the different water populations in the cementitious porous network. The TGA measurements showed that both CNFs and 3D printing influenced the different water populations in cement composites and, consequently, the porous network of the material. With regard to the electrical properties investigated by dielectric spectroscopy, a characteristic behavior as a function of temperature can be observed in all samples, consistent with the ranges found by TGA in relation to the loss of different water populations. We can thus assume that the increase in σ(ω) is mainly determined by $c_{W}^{I}$, i.e., by the degassing of water at T < 80 ${}^{\circ }C$. When all the evaporable water was removed, a decrease in conductivity was observed. This result not only underlines the importance of water as a medium for electrical conductivity in cementitious materials, but also shows that each different water population plays a different role in such a process. Consequently, changes in the porous network led to variations in the distribution and quantity of the different water populations and, thus, to changes in the conductive properties of the material. To investigate the existence of a clear dependence of σ(ω) with the different water contents, Figure 5 shows the electrical conductivity as a function of $c_{w}$. A clear linear dependence of σ(ω) with $c_{w}^{I}$ was observed. Such behavior was not found in the other water populations, suggesting that it is typical of water outgassing at low temperatures. Furthermore, the orange dot in Figure 5a is the value of the conductivity of the sample prepared by the traditional method without conductive fillers. The value fell out of the linear trend found in the 3D-printed samples, indicating a strong correlation with the sample preparation technique. A similar linear dependence was found in the work of Liu et al. on the study of electrical properties in cement slurries for free water, which is related to the large and interconnected pores in the cementitious matrix [56]. Moreover, tomography of 3D-printed samples in the work of Kruger et al. [57] showed strong differences in the porous network compared to that produced by the traditional method. In fact, the pores lost their spherical shape during 3D printing, and likely interconnections were shown. Moreover, it is not possible to observe a clear relationship between the $c_{w}^{I}$ content and the CNF dosage (see Figure 6), i.e., the addition of conductive filler did not directly affect the amount of water outgassing at low temperature. Therefore, we can assume that the $c_{w}^{I}$ content is related to the free water in the interconnected pores, which are mainly a consequence of the 3D printing technique. Finally, Figure 7 shows the electrical conductivity as a function of dosage at room temperature (a) and at 160 ${}^{\circ }C$ (b). Two different behaviors were observed depending on the temperature range. At room temperature, it is not possible to see a clear correlation between the carbon fillers and the conductivity. Consequently, in this temperature range, the dosage of CNF had no direct influence on the conductivity of the cement composites. At high temperatures (T = 160 ${}^{\circ }C$), on the other hand, a different dependence was observed. In samples with CNF dosages of less than 4 wt%, a sharp decrease in conductivity of 1–2 orders of magnitude was observed, confirming that, in these samples, the main contribution to conductivity was due to charge carriers displaced by water in the porous network. In contrast, the sample with 4 wt% of CNF showed only a slight decrease in σ(ω) after annealing at a high temperature. This behavior can be explained by the assumption that a conductive pathway formed. Such a result is somewhat at odds with previous work in which a percolation threshold was found for a smaller amount of carbon filler. To understand this discrepancy, we can assume that the CNFs dispersed in the cementitious matrix act as a bridge between adjacent pores [58]. The presence of water molecules in the porous system helps to create a continuous conduction path [59], and when the water evaporates, this path is interrupted [35]. Since, in our work, the electrical conductivity in the sample with 4wt% of CNF did not depend on the amount of water in the porous network, we assumed that a true conductive path formed, while the percolation value obtained with a low dosage of conductive filler was related to the path formed by the CNFs bridging the porous network.

## 5. Conclusions

In this work, we explored the effects of 3D printing on the electrically conductive properties of concrete composites with CNFs. First, an additional contribution to the electrical performance was observed once the 3D printing process was used for sample preparation. In this work, the orientation of the carbon nanofibers was apparently not directly associated with such an effect. TGA measurements revealed that both 3D printing and CNF dosing influenced the distribution of water molecules in the cementitious porous network. Specifically, the CNF concentration affected the water molecules interacting with the host, while 3D printing affected the structural amount of water, suggesting a higher degree of hydration of the samples. Finally, two different water populations were observed in the bulk.

Regarding the dependence of electrical conductivity on temperature, a characteristic behavior was observed in the 3D-printed specimens. The temperature range defining the different trends was consistent with the range observed in the TGA, highlighting the strong relationship between water populations and electrically conductive behavior. Furthermore, a linear dependence between σ(ω) and water outgassing at low temperatures was observed. Since the content of this water population does not depend on the dosage of CNFs, we can assume that this effect is mainly due to the 3D printing process. We suggest that the use of the extruder changed the porous network by creating more interconnected large pores, and the free water molecules in these channels controlled the electrical conductivity at room temperature. On the other hand, the study showed a slight decrease in σ(ω) at high temperatures once the sample was heated to 160 ${}^{\circ }C$ with 4 wt% of CNF, suggesting the presence of a complete conduction pathway. 

## Figures and Tables

**Figure 1 nanomaterials-12-03939-f001:**
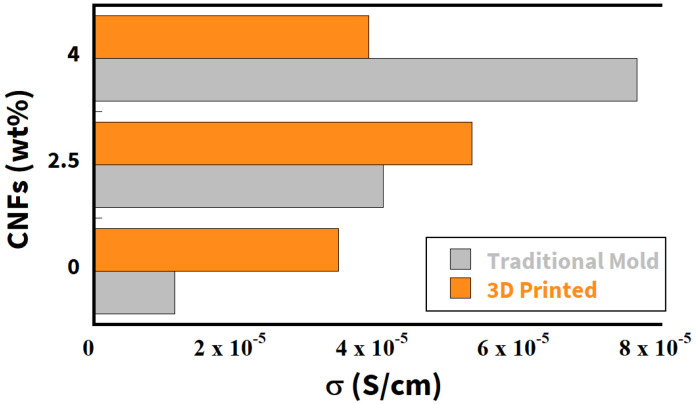
Comparison of electrical conductivity measured by dielectric spectroscopy at RT and at 100 Hz between samples prepared with the traditional casting mold procedure (gray) and 3D printing (orange) as a function of CNF dosage.

**Figure 2 nanomaterials-12-03939-f002:**
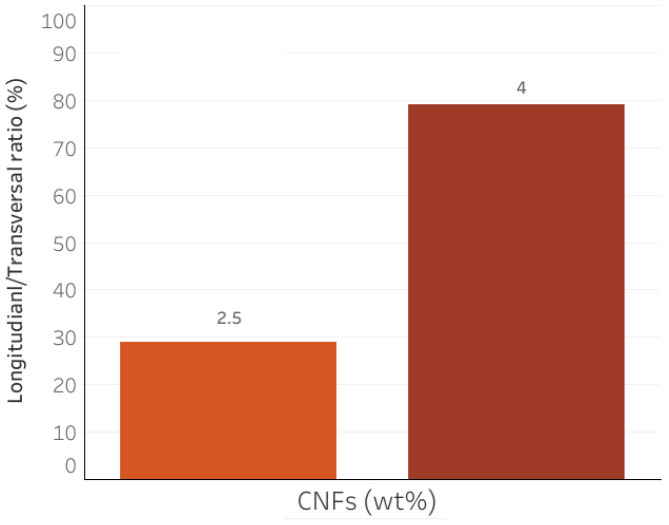
Electrical conductivity ratio of the electrical conductivity measured along the longitudinal/transversal direction of extrusion.

**Figure 3 nanomaterials-12-03939-f003:**
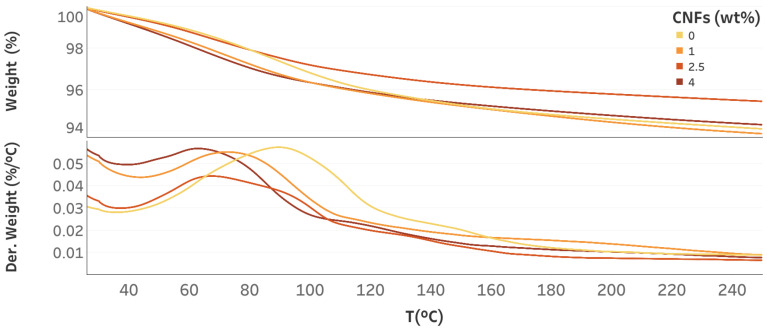
TGA curves: weight loss % (top) and derivative weight loss ($\%{/}^{\circ }C$) (bottom) of specimens with different CNF dosages after 7 days at RH = 77.

**Figure 4 nanomaterials-12-03939-f004:**
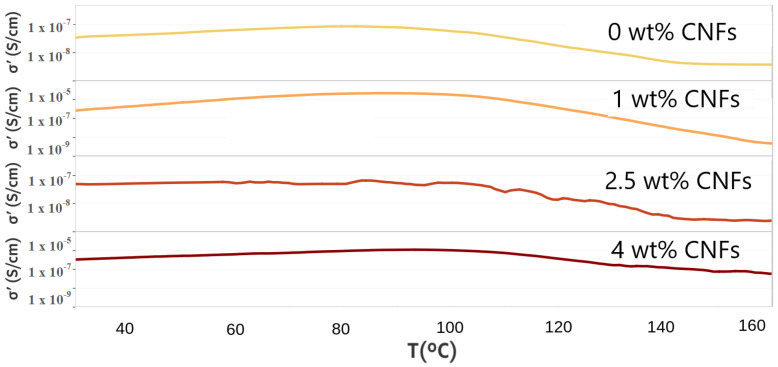
Real part of the dielectric complex function measured at 100 Hz as a function of T.

**Figure 5 nanomaterials-12-03939-f005:**
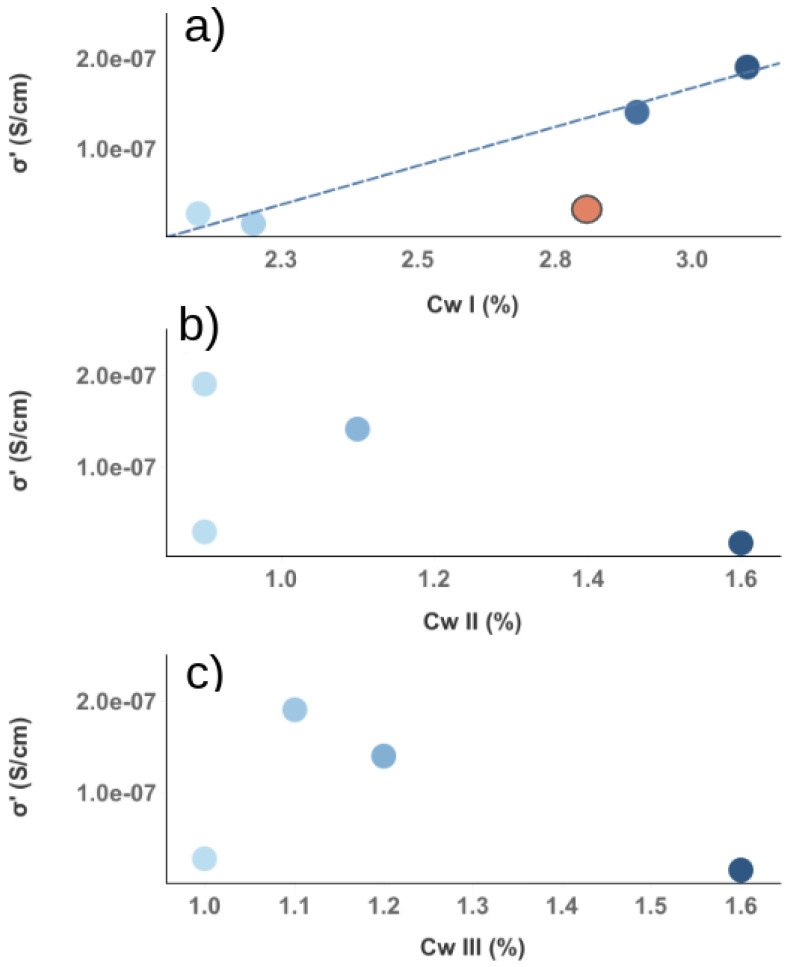
Electrical conductivity values at 100 Hz as a function of water population content. Gradient from light to dark blue refers to water content in 3D printed sample. In (**a**), the dashed line is a visual guide to highlight the linear dependence of conductivity with water population outgassing at a low temperature. In contrast, it is impossible to observe any dependence between conductivity and water amount considering the water population outgassing between 80 and 105 ${}^{\circ }C$ (**b**) and the strongly bound water (**c**).

**Figure 6 nanomaterials-12-03939-f006:**
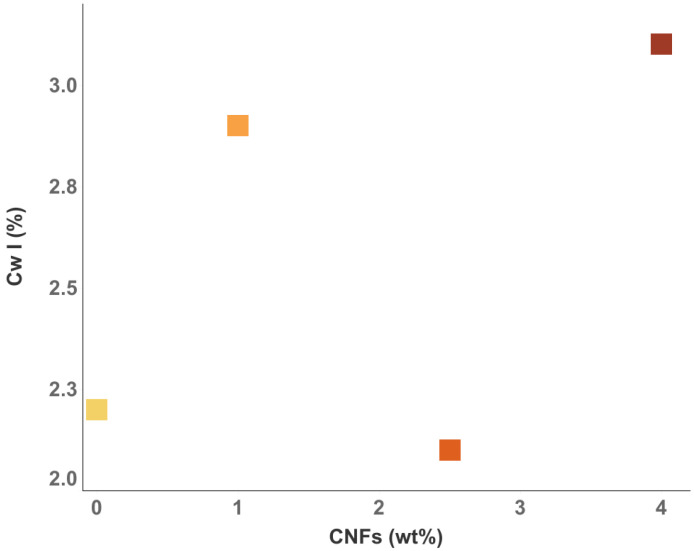
Content of $c_{W}^{I}$ water population as a function of CNF wt% (Gradient from yellow to dark red refers to CNFs amount).

**Figure 7 nanomaterials-12-03939-f007:**
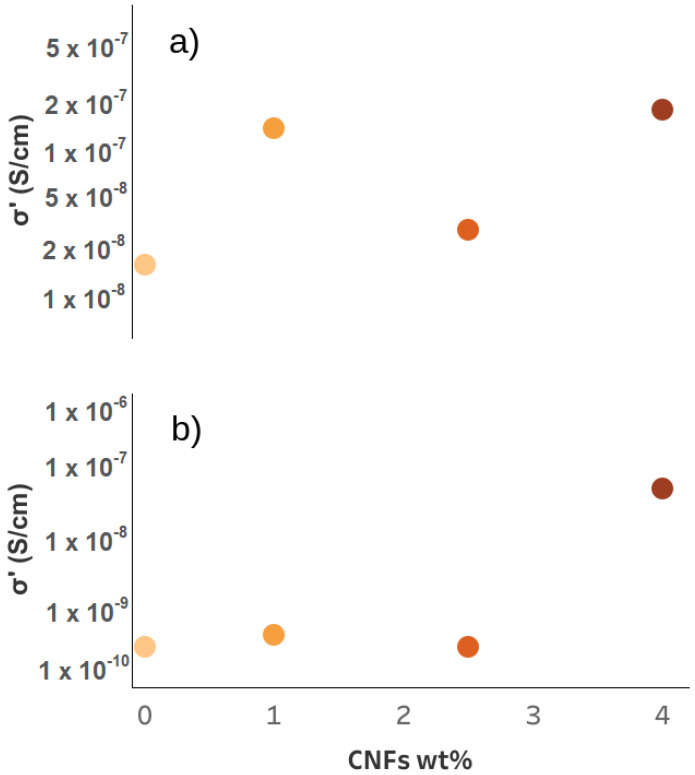
Electrical conductivity at 100 Hz as a function of CNF dosage at 30 ${}^{\circ }C$ (**a**) and 160 ${}^{\circ }C$ (**b**) (Gradient from yellow to dark red refers to CNFs amount).

**Table 1 nanomaterials-12-03939-t001:** Mixture design of electrical conductive concretes.

Sample	Cem I-52.5R (g)	Limestone (g)	Water (g)	CNFs (g)
0	450	1350	270	0
1	450	1350	270	4.5
2.5	450	1350	270	11.25
4	450	1350	270	18

**Table 2 nanomaterials-12-03939-t002:** Water content (%) of the distinct water populations confined in the specimen obtained by TGA measurements.

CNFs (wt%)	$c_{W}^{\mathrm{TOT}}\left(\%\right)$	$c_{W}^{I}$ (%)	$c_{W}^{\mathrm{II}}$ (%)	$c_{W}^{\mathrm{III}}$ (%)	$c_{W}^{\mathrm{IV}}$ (%)
0 (Traditional Casting)	5.5	2.9	1.4	1.0	1.2
0	6.3	2.2	1.6	1.6	2.5
1	6.5	2.9	1.1	1.2	2.5
2.5	4.8	2.1	0.9	1.0	1.8
4	6.0	3.1	0.9	1.1	2.0

## Data Availability

Not applicable.

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
