# Peer review of "Electrical Conductive Properties of 3D-Printed Concrete Composite with Carbon Nanofibers"

_nanomaterials, 2022, doi:10.3390/nano12223939_

Round 1
Reviewer 1 Report
Comments to the Authors
In this manuscript authors investigated the conductive performance of cement composites with carbon nanofibers by means of a combination of thermogravimetric analysis and dielectric spectroscopy. This research has value for the researchers in the related areas. However, the paper needs improvement before acceptance for publication. My detailed comments are as follow:
1. In the introduction section authors should use the following carbon nanofibers related article as references and their discussion :
a. Bhawal, P., et al. "Fabrication of light weight mechanically robust short carbon fiber/ethylene methyl acrylate polymeric nanocomposite for effective electromagnetic interference shielding." J Polym Sci Appl 1 (2017): 2.
b. doi.org/10.1007/s42114-018-0072-z
2. In the Figure 2 authors should provide error bar.
3. The writing of abstract section should be improved.
4. In Table 1 caption is missing.
5. In figure 3 Y axis label should be corrected.
6. In a small section authors should discuss what will be effects if the orientation is random?
7. There are few grammatical errors authors should correct it.
Author Response
In this manuscript authors investigated the conductive performance of cement composites with carbon nanofibers by means of a combination of thermogravimetric analysis and dielectric spectroscopy. This research has value for the researchers in the related areas. However, the paper needs improvement before acceptance for publication. My detailed comments are as follow:
- In the introduction section authors should use the following carbon nanofibers related article as references and their discussion :
- Bhawal, P., et al. "Fabrication of light weight mechanically robust short carbon fiber/ethylene methyl acrylate polymeric nanocomposite for effective electromagnetic interference shielding." J Polym Sci Appl 1 (2017): 2.
- doi.org/10.1007/s42114-018-0072-z
We thank the reviewer for the suggestion: the reference has been added to the manuscript (line 20).
- In the Figure 2 authors should provide error bar.
Figure 2 refers to the ratio in % between conductivity measured along the longitudinal and transverse directions of the 3D printing. Consequently, an error bar cannot be provided. However, the misunderstanding may be given by an unclear description in both capture of the figure and text of the manuscript. Therefore, some sentences have been rephrased.(line 112)
- The writing of abstract section should be improved.
The referee is right: we have made changes to improve the quality of the abstract
- In Table 1 caption is missing.
We are sorry for the inconvenience. Caption has been added to table 1.
- In figure 3 Y axis label should be corrected.
For figure 3 we used the TGA standard label. However, a better introduction of the labels has been added to the caption of the figure.
- In a small section authors should discuss what will be effects if the orientation is random?
The referee is right. A short description of the expected effect of a random orientation of CNFs was has been added (line 113).
- There are few grammatical errors authors should correct it.
We thank the reviewer for the comment. The article has been deeply revised.
Reviewer 2 Report
The paper "Electrical conductive properties of 3D printed concrete composite
with carbon nanofibers" from Goracci et al. deals with the investigation of the conductivity in carbon/concrete composites and how it is affected by the presence of filler, water and by the preparation method.
The goal of this study is interesting and worth investigating, and the approach employed is valid. The manuscript is well structured but could use some refinement in terms of the language employed. There are several typos and grammar mistakes that could be fixed with the help of a native english speaker. The results are of interest for the scientific community and for industries working on the development and exploitation of new cement and concrete materials. The conclusions are reasonable and are supported by the experimental results
From my point of view there is only one main concern with this paper, but is a crucial one: there is no mention on the number of samples investigated for each composition, and if the same samples were measured with the different techniques. The conclusion are perfectly reasonable if the investigation is statistically sound, but if the variation of the properties for the same composition and preparation method is large the situation might be entirely different. For these kind of samples this would not be unreasonable and should be verified. This kind of investigation should be reported and explained in the text.
Minor suggestions:
-page1: CB not defined
-page16 line 116: the implications of the last sentence of the section are only evident after the discussion in page 7, it would help the reader if a hint was given also at this stage.
-page 3: RH not defined
-page4: missing caption of table 1
-pages 4-5: the same unit for temperature should be used in fig 3 and 4
Author Response
The paper "Electrical conductive properties of 3D printed concrete compositewith carbon nanofibers" from Goracci et al. deals with the investigation of the conductivity in carbon/concrete composites and how it is affected by the presence of filler, water and by the preparation method.
The goal of this study is interesting and worth investigating, and the approach employed is valid. The manuscript is well structured but could use some refinement in terms of the language employed. There are several typos and grammar mistakes that could be fixed with the help of a native english speaker. The results are of interest for the scientific community and for industries working on the development and exploitation of new cement and concrete materials. The conclusions are reasonable and are supported by the experimental results
From my point of view there is only one main concern with this paper, but is a crucial one: there is no mention on the number of samples investigated for each composition, and if the same samples were measured with the different techniques. The conclusion are perfectly reasonable if the investigation is statistically sound, but if the variation of the properties for the same composition and preparation method is large the situation might be entirely different. For these kind of samples this would not be unreasonable and should be verified. This kind of investigation should be reported and explained in the text.
The reviewer is right. A better explanation of the experimental method used is necessary to understand the possible variation in the properties observed. We measured one sample per dosage. The sample investigated was taken from a large extruded specimen (1m length) and it was cut always at the same position. To improve the correlation between the TGA and DS experiments, a few milligrams of the samples were removed for the TGA analysis and both experiments were performed on the very same day. This information has been added to "Materials and Methods" section.
Minor suggestions:
-page1: CB not defined
We thank the reviewer for the comment, CB was defined in line 30.
-page16 line 116: the implications of the last sentence of the section are only evident after the discussion in page 7, it would help the reader if a hint was given also at this stage.
Sentence in line 116 is referred to Fig 1 (measurement at room temperature before keeping the specimen at relative humidity RH77). We added a short comment in the text to remove possible misunderstanding (line 127)
-page 3: RH not defined
The reviewers is right, RH has been correctly defined in line 133.
-page4: missing caption of table 1
We are sorry for the inconvenience. Caption has been added to table 1.
-pages 4-5: the same unit for temperature should be used in fig 3 and 4
Fig 4 was modified to have the same temperature unit (celsius). Consequently, even in text only Celsius has been used.
Reviewer 3 Report
In this manuscript, the authors used 3D printing technology to print carbon nanofibers -modified concrete composite and investigate the on the electrical conductive properties of concrete composites with CNFs. However, the designed experiment and the data analysis were too rough and did not follow scientific investigation logic. Therefore, this publication cannot be accepted by Nanomaterials.
1. In the introduction part, it is said “but also the enhancement of secondary properties, such as electrical conductivity, to tailor the composite for the desired function” (Line 19). Could you please the scenarios using concrete/CNF composite exhibiting 6e-5S/cm conductivity. I still wonder the necessity to add conductive materials in concrete. In Line 21, it is said ” In this work, we want to add one more piece to the electrically conductive investigation exploring how 3D concrete printing (3DCP) can affect the conductive properties of concrete composite.” Why 3D printing is needed to fabricate conductive concrete?
2. The rheological performance (viscosity, elastic modulus, viscous modulus) and the optical images should be provided to support the introduction of these 3D printed samples.
3. The author mentioned about the alignment of CNF when discussing the samples’ electrical conductivity difference. Thus, SEM images are expected to verify the assumption. And analysis based on relationship between size of nozzle diameter and CNF length could be discussed to have more insightful understanding of the alignment behavior. Also, Figure 2 should have more CNF concentration variables than just two, which is inadequate for having conclusions.
4. It seems the water’s effects on conductivity is straightforward and don’t need to be investigated in such complicated way.
Other comments:
1. The data in the manuscript should be adjusted to follow the scientific standard for the reader to easily understand. For example, Figure 1 could be changed to a bar graph with vertical and horizontal axes. Figure 2 should provide the information of horizontal axis.
2. Some of the figures are not clear, for example Figure 5.
3. Figure 1, it is said ” with the traditional casting mold procedure (right) and 3D printing (right) as a function of CNFs dosage”. One of the “right” should be left.
4. Table 1 should be placed on the top of the table. And the title should provide detailed information instead of using “Table 1 Caption”.
5. The Materials and Methods section could be modified to a more organized way.
6. It is suggested not divide “Result and discussion“ in two section, which makes the description repeated and logic interrupted.

Author Response
In this manuscript, the authors used 3D printing technology to print carbon nanofibers -modified concrete composite and investigate the on the electrical conductive properties of concrete composites with CNFs. However, the designed experiment and the data analysis were too rough and did not follow scientific investigation logic. Therefore, this publication cannot be accepted by Nanomaterials.
1. In the introduction part, it is said “but also the enhancement of secondary properties, such as electrical conductivity, to tailor the composite for the desired function” (Line 19). Could you please the scenarios using concrete/CNF composite exhibiting 6e-5S/cm conductivity. I still wonder the necessity to add conductive materials in concrete. In Line 21, it is said ” In this work, we want to add one more piece to the electrically conductive investigation exploring how 3D concrete printing (3DCP) can affect the conductive properties of concrete composite.” Why 3D printing is needed to fabricate conductive concrete?
We thank the reviewer for the comment.
“Could you please the scenarios using concrete/CNF composite exhibiting 6e-5S/cm conductivity”
We are aware that the conductivity reached in our work is still low for many applications (even though values fall in the range accepted for electrical conductive concretes as reported in “A review on material design , performance , and practical application of electrically conductive cementitious composites” Wang et al., Construction and Building Materials 2019,). However, the scope of our work is not to propose a novel material for applications, but to explore the use of a novel construction process.
“I still wonder the necessity to add conductive materials in concrete”
As reported in the introduction, there are many applications where the electrical conductive properties of concrete composites are fundamental. We decided to add a further and recently investigated application of electrical conductive concrete, that is, thermoelectric concrete for energy harvesting. In this application the low conductivity is the main drawback and the improvement of such a property is fundamental for a real application.
“ Why 3D printing is needed to fabricate conductive concrete”
This is the most important finding of this paper, namely that 3D printing can affect the effective electrical conductivity of concrete.
2. The rheological performance (viscosity, elastic modulus, viscous modulus) and the optical images should be provided to support the introduction of these 3D printed samples.
The reviewer is right, rheological performance is critical in 3D printing of concrete. Workability and viscosity are relevant parameters during the extrusion process. However, we consider that such an investigation is out of the scope of the work presented. In fact, we focused our attention on the conductive properties and their relation with water populations to explore the use of 3D printing for electrical conductive concrete composites, since this technique has been poorly investigated for this aim.
3. The author mentioned about the alignment of CNF when discussing the samples’ electrical conductivity difference. Thus, SEM images are expected to verify the assumption. And analysis based on relationship between size of nozzle diameter and CNF length could be discussed to have more insightful understanding of the alignment behavior. Also, Figure 2 should have more CNF concentration variables than just two, which is inadequate for having conclusions.
We agree with the reviewer that SEM pictures could confirm the alignment, although we also notice that the study would be tremendously complex and barely conclusive. The nanofibers are “nano” and only with high magnification would be possible detection. Besides, the SEM collects 2D information where the 1D character of the nanofibers makes them quite elusive for detection.
Finally, Figure 2 does not want to highlight conclusions on the allignement. Its function is to show that, in our work, we have found that the sample with higher alignment has a lower electrical conductive performance, suggesting that the additional contribution to conductivity must reside somewhere else.
4. It seems the water’s effects on conductivity is straightforward and don’t need to be investigated in such complicated way.
We disagree with the comment of the reviewer. The relationship between water and conductivity may appear straightforward. However, due to the complex porous network of cementitious materials, the distribution of different water populations (each characterised by different dynamical and structural behaviour) is a nontrivial topic that must be investigated. New experimental methods must be used to reveal the relation between the water populations and the macroproperties of the material (thermal, structural and, of course, electronic behaviour). A deeper knowledge of this topic will help design the best mixture for each application.
Other comments:
1. The data in the manuscript should be adjusted to follow the scientific standard for the reader to easily understand. For example, Figure 1 could be changed to a bar graph with vertical and horizontal axes. Figure 2 should provide the information of horizontal axis.
We thank the reviewer for the comment. Figure 1 and 2 have been modified
2. Some of the figures are not clear, for example Figure 5.
We improved the comment in the capture to describe better the figures.
3. Figure 1, it is said ” with the traditional casting mold procedure (right) and 3D printing (right) as a function of CNFs dosage”. One of the “right” should be left.
The reviewer is right. The error has been corrected.
4. Table 1 should be placed on the top of the table. And the title should provide detailed information instead of using “Table 1 Caption”.
We are sorry for the inconvenience. Caption has been added to table 1.
5. The Materials and Methods section could be modified to a more organized way.
The reviewer is right. The Materials and Method section was re-organized and improved.
6. It is suggested not divide “Result and discussion“ in two section, which makes the description repeated and logic interrupted.
We thank the reviewer for the comment. However, we consider that maintaining Results and Discussion in two separate sections is the best option. To overcome the repetition of unnecessary descriptions, the first part of the discussion was shortened.
Round 2
Reviewer 2 Report
The authors did not resolve the main issue presented in my previous report, and confirmed that they did not collect any statistics on the samples. Therefore the conclusions they draw might be coming from just fluctuations of the properties in such an intrinsically inohomogeneous material. The authors state that the small measured samples were taken from large extruded specimen. To confirm the validity of their results they should carry out the measurements on a few more samples obtained from the same specimen for at least one of the compositions. If the variation of the properties for different composition exceeds the variations within the same composition, then the results and conclusions are confirmed. An alternative possibility would be to make sure that the already analyzed samples do not have any significant variations of compositions, density, etc other than those subject of the study.
The rest of the paper is well prepared, but this issue in my opinion can not be avoided.
Reviewer 3 Report
The authors‘ response letter solves most of the proposed questions.